# Mars Exploration Using Sailplanes

Adrien Bouskela [1], Alexandre Kling [2], Tristan Schuler [1], Sergey Shkarayev [1], Himangshu Kalita [1] and Jekan Thangavelautham [1,*]

1   Department of Aerospace and Mechanical Engineering, University of Arizona, Tucson, AZ 85721, USA; adrienbouskela@email.arizona.edu (A.B.); tristan.schuler@nrl.navy.mil (T.S.); svs@email.arizona.edu (S.S.); hkalita@email.arizona.edu (H.K.)
2   NASA Ames Research Center, Mountain View, CA 94043, USA; alexandre.m.kling@nasa.gov
*   Correspondence: jekan@arizona.edu

**Abstract:** We present the preliminary design of sailplanes, used for Mars exploration. The sailplanes mitigate the weight and energy storage limitations traditionally associated with powered flight by instead exploiting atmospheric wind gradients for dynamic soaring, and slope/thermal updrafts for static soaring. Equations of motion for the sailplanes were combined with wind profiles from the Mars Regional Atmospheric Modeling System (MRAMS) for two representative sites: Jezero crater, Perseverance's landing site, and over a section of the Valles Marineris canyon. Optimal flight trajectories were obtained from the constrained optimization problem, using the lift coefficient and the roll angle as control parameters. Numerical results for complete dynamic soaring cycles demonstrated that the total sailplane energy at the end of a soaring cycle increases by 6.8–11%. The absence of a propulsion system, allowing for a compact form factor, means the sailplanes can be packaged into CubeSats and deployed as secondary payloads at a relatively low cost; providing scientific data over locations inaccessible by current landers and rovers. Various sailplane deployment methods are considered, including rapid deployment during Entry, Descent, and Landing (EDL) of a Mars Science Laboratory-class (MSL) vehicle and slow deployment using a blimp.

**Keywords:** Mars; sailplane; CubeSat; dynamic soaring; atmosphere; blimp





## 1. Introduction

Exploration of terrestrial planets, such as Mars, are currently conducted using orbiters, landers, and rovers. Two outstanding areas of research that could benefit from aerial platforms are the characterization of past habitability (e.g., identifying the nature, ages, and origin of the diverse suite of geologic units), which is supported by mapping and the characterization of the current physical processes at play in the Martian atmosphere, near the surface (e.g., changes in the wind and temperature structures, transport of dust, ice aerosols, and water vapor) [1,2].

Cameras and instruments onboard orbiters have enabled global mapping of Mars at resolution of approximately 0.3 m/pixel. Landers and rovers, such as the MSL and Perseverance, carry state-of-the-art instruments to extensively characterize small local areas with even greater mapping resolution [3,4]. However, this leaves large tracts of Mars largely unexplored except for on-orbit imagery. There is a critical gap in exploration capabilities—mapping a mesoscale view of regions with less than 0.1 m/pixel resolution over hundreds of kilometers. Similarly, remote sensing instruments aboard orbiters allow for retrieval of temperature and aerosol structures, but cannot probe within a few km of the surface [5], which is a critical knowledge gap an aerial platform would fill.

A large number of sites of interests on Mars remain inaccessible due to current technological limitations in precision EDL and the inability to land in high-altitude and rugged terrains. Previously, several airborne missions providing access to these remote sites have been proposed for Mars exploration [6–10]. Most of the previous concepts are based on

powered flying vehicles and the inclusion of electric or chemical propulsion would produce significant penalty on the mass, complexity and reliability of the aircraft.

To address the challenges of long duration flight (few hours to a few days) with limited resources, nature provides an elegant source of inspiration in the form of the albatross, a bird able to sustain flight for thousands of miles over the ocean without landing. The secret of albatross's flight lies in utilizing its environment through dynamic soaring maneuvers [11,12].

The idea of a heavier-than-air aircraft for Mars exploration holds much promise for future discoveries on the planet. A small helicopter designed by NASA JPL [13] launched onboard Mars Perseverance Rover in 2020 and performed its first flight on another planet. The helicopter has two counter-rotating rotors and has a total mass of 1.8 kg. Flights performed were on the order of 2 min each [14].

The autonomous aircraft ARES (Aerial Regional-scale Environmental Survey) was proposed by NASA Langley as a Discovery mission candidate [15]. Several innovative propulsion technologies were considered for powering this vehicle, including electrical motors, internal combustion and rocket systems suited for Mars exploration. The aircraft was intended to fly about an hour performing atmospheric probing missions. The lack of sufficient atmospheric data and models deemed the mission too risky compared to its competitors, such as the University of Arizona led Mars Phoenix mission that was to look for water-ice near the Martian North Pole. An unconventional craft called Entomopter was proposed in the previous study [16]. It uses two sets of flapping wings mounted on the fuselage. The wings generate insect-like motions that produce a lift and thrust force. The flight endurance is predicted to be 10–15 min. Note that the flight endurance of the aircraft proposed in the previous studies have been limited by the amount of fuel/energy carried on board. Alternatively, the unpowered glider, Prandtl-m was designed [17], based on the high-aspect-ratio flying wing concept. It is foldable and fits into a CubeSat. However, the glider is capable of only about 10 min of flight time.

Among limitations and challenges of the previous research and development efforts, the most critical one has been short flight endurance. To overcome this underperformance, we propose to design an unpowered sailplane that employs a multitude of dynamic and static soaring methods for flight in the Martian atmosphere. The Mars Sailplane has the capability of mapping a mesoscale view of regions with less than 0.1 m/pixel resolution ranging in tens to hundreds of kilometers and addresses current platform limitations. The aircraft can dive to tens of meters altitude to obtain even higher resolution images of a particular feature or area. This sailplane concept, unlike previous proposals for Mars, would be a secondary-payload, and, thus, low-cost. We estimate the sailplane mission costing $100 million or less, as opposed to Discovery or a flagship mission that would be $300 million or more. A defining advantage of this sailplane concept over previous proposals is its simplicity. The system does not require an engine that imposes significant penalty on mass, complexity, and reliability to take flight. In addition, an engine to power flight on Mars will require overcoming significant development hurdles, particularly for the engine to utilize the thin, dusty Martian $CO_2$ atmosphere to provide sustained and reliable thrust.

Dynamic soaring of sailplanes in the Earth's atmospheric boundary layer has been well studied by using simulations and flight tests. Analysis of work-energy relationship of a sailplane in wind is presented in [18]. The study shows that the energy neutral cycle depends on the maximum lift-to-drag ratio of the vehicle and the wind speed gradient. For a continuous wind profile, the minimum required gradient of the wind was calculated, providing the neutral energy cycle.

The dynamic soaring maneuvers of unmanned aerial vehicles in the atmospheric boundary layer were examined numerically for several different wind profiles [19]. Both 3-DOF and 6-DOF models were employed [20] to demonstrate dynamic soaring with extreme climbs to high-altitudes in high wind conditions. The flight demonstrations on Earth [21,22] have proven the effectiveness of these maneuvers in achieving long-endurance flights.

In our previous work [23], we explored the initial feasibility of sailplane deployment and dynamic soaring in the Martian atmosphere. We found that the lower gravity and atmospheric pressure do not hinder an aircraft's ability to perform required energy-gaining maneuvers, and that higher flight speed, and consequentially higher distances, are necessary due to the low Reynolds number conditions.

Our current work focuses on flying over canyons and craters using static soaring and dynamic soaring methods. We employed point-mass simulations to study flight trajectories in deep canyons. The simulations were conducted using the Mars Regional Atmospheric Modeling System (MRAMS) [24], which provides new, realistic conditions for dynamic soaring on Mars. Several deployment strategies were explored, including quick deployment during EDL and using a carrier hot-air balloon or blimp.

## 2. Sailplane Mission for Mars Exploration

A notional Mars sailplane mission consists of five successive phases: (1) the initial deployment and leveling, (2) direct navigation to science target by using large-scale currents, balloons, and blimps, (3) search, location, and observation of a science target with multiple passes, (4) station keeping and long-distance navigation using soaring maneuvers to secondary targets, and finally (5) landing and extending surface meteorology operations.

Figure 1 describes possible flight patterns over a canyon of two sailplanes S1 and S2. The sailplane S1 would deploy and navigate to the canyon floor, perform sustained flight and science imaging using ground proximity dynamic soaring, either advancing down the canyon or loitering over local targets of interest. Images and wind measurements collected in previous flights will provide insight for a second sailplane S2. It would arrive to the area at a later local time to fly high amplitude slope soaring trajectories along the canyon wall, surveying the cliff sides with multiple climbing and descending passes, finishing with enough energy for a secondary target. This assumes that favorable atmospheric conditions (winds) allow for gliding along the wall face for the entire length of the canyon. Each sailplane terminates its primary mission with a soft belly landing on the Martian surface and transforms into a meteorological station. The advantage of this approach is that meteorological stations can be dispersed throughout the surface of Mars without requiring additional maneuvering of the EDL.

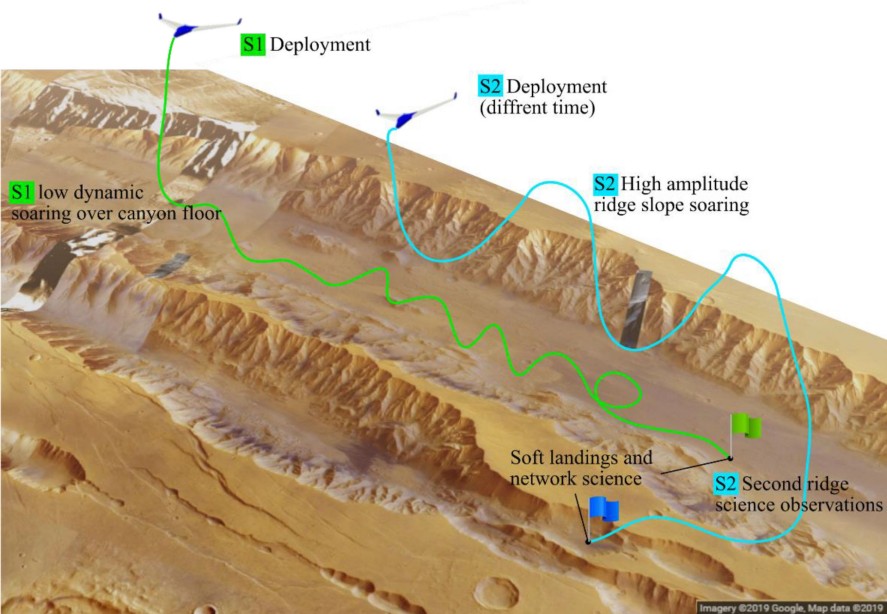

**Figure 1.** Concept of operations of sailplanes S1 and S2.

## 3. Aerodynamic Design of Sailplane

The Martian atmosphere is a hundred times thinner than that of Earth ($\rho = 0.0137 \, \mathrm{kg/m^3}$). Even though gravity on Mars is significantly lower, $g = 3.727 \, \mathrm{m/s^2}$, the combination makes sustainable flight challenging on this planet. With a viscosity of $\mu = 1.08 \times 10^{-5} \, \mathrm{N \, s/m^2}$, a craft of a mass of 5 kg, with an assumed wingspan of 3.35 m and mean aerodynamic chord of 0.54 m, has a speed of 70 m/s and the Reynolds number of 47,950. This value falls within the low Reynolds number range (low Re is less than $2 \times 10^5$). In contrast, the Mach number of high-speed flights can reach 0.8. While a limited number of low Reynolds number airfoils have been studied [25–27], not much research has been conducted for a combination of moderate Mach and low Reynolds number conditions.

Design problems associated with low atmospheric density and low Reynolds number have been considered in the present work. A number of thin airfoils [25] were analyzed and the airfoil S9033 was selected, due to its high lift-to-drag ratio and moderate pitching moment coefficient.

A sweptback flying wing design is proposed that provides better performance in comparison to a conventional fixed-wing aircraft with a tail. The sailplane is shown in Figure 2. A notable feature of this configuration is the lift-generating fuselage, which is seamlessly blended with the body (blended-wing-body design). Not only is the lift produced by the entire craft increased, but the interference drag is reduced. To determine the best parameters of this design, we conducted an optimization of the aerodynamic efficiency in our previous study [28]. The specifications of the obtained sailplane design are given in Table 1.

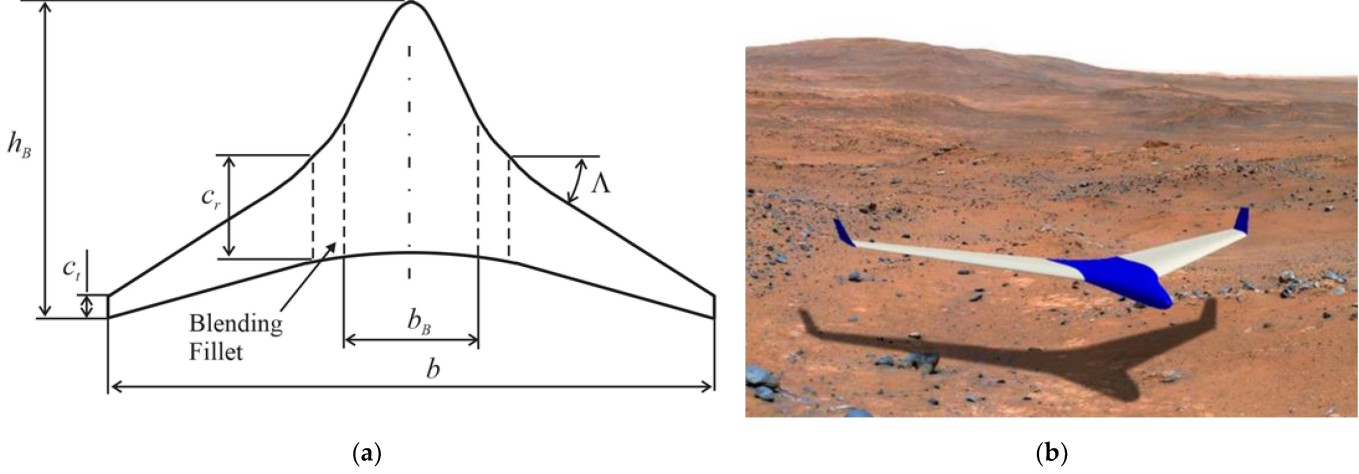

(a)              (b)

**Figure 2.** (**a**) Sailplane platform; (**b**) 3D-rendering.

**Table 1.** Specifications of Mars Sailplane.

| Parameter | Value | Parameter | Value |
|---|---|---|---|
| Wing/Body Airfoil | S9033 | Wing Area, $[\mathrm{m^2}]$ | 1.8 |
| $b$, $[\mathrm{m}]$ | 3.35 | Aspect Ratio | 6.2 |
| $b_B$, $[\mathrm{m}]$ | 0.7 | Average Speed, $[\mathrm{m/s}]$ | 70 |
| $c_r$, $[\mathrm{m}]$ | 0.56 | Minimum Sink, $V_{sink}$, $[\mathrm{m/s}]$ | 6.28 |
| $c_t$, $[\mathrm{m}]$ | 0.34 | Stall Speed, $V_{stall}$, $[\mathrm{m/s}]$ | 42 |
| $h_B$, $[\mathrm{m}]$ | 1.12 | Total Mass, $[\mathrm{kg}]$ | 5.0 |
| $\Lambda$, $[\mathrm{deg}]$ | 45 | Flight Endurance | 20 min to days |

## 4. Numerical Model for Flight in Martian Atmosphere

Specific flights will be adapted to specific mission requirements on Mars, e.g., low pass flight over a particular site. They will include the following flight patterns: dynamic soaring, gliding, soaring in updrafts, and dive and climbing flights.

Consider a flight path of a sailplane modeled as a point-mass, m, with three degrees of freedom. Figure 3 illustrates conventions for forces, angles, and velocities. There are three applied forces: lift, $L$, drag, $D$, and gravitational force, mg. The aerodynamic lift and drag are conventionally presented as:

$$L = 0.5C_L\rho V_a^2 S; D = 0.5C_D\rho V_a^2 S \tag{1}$$

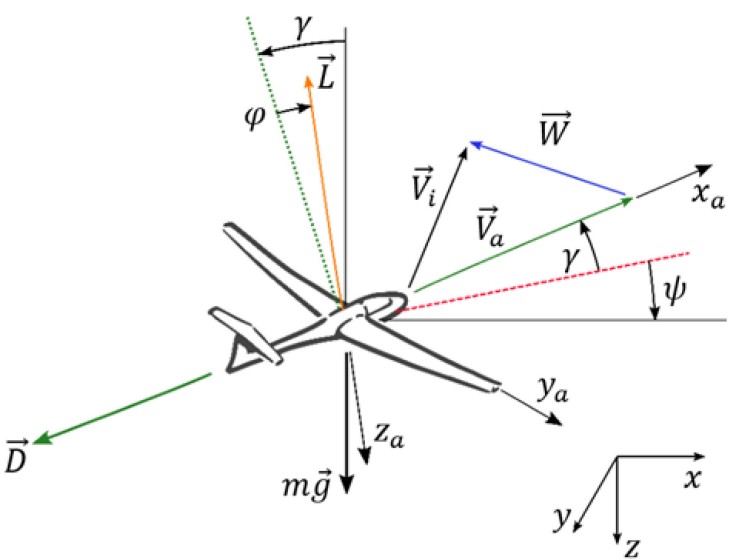

**Figure 3.** Conventions for forces, angles, and velocities.

The drag polar for the flying wing with the S9033 airfoil was calculated with the help of the program [29] and the lift coefficient is bound as $C_L \in \left[C_{L_{min}}, C_{L_{max}}\right]$.

The sailplane flight is described using an inertial reference frame $(x, y, z)$, and the aerodynamic frame of reference $(x_a, y_a, z_a)$ attached to the sailplane. The transformation from the inertial to the aerodynamic frame is performed using three canonical rotations about the yaw angle, $\psi$, the pitch, $\gamma$, and the roll, $\varphi$, which are Euler angles. The components of the velocity vector of the vehicle, relative to atmosphere, are:

$$\vec{V_a} = V_a[cos(\psi)cos(\gamma), cos(\gamma)sin(\psi), -sin(\gamma)]^T \tag{2}$$

and components of wind in the inertial frame are denoted as $\vec{W} = [uvw]^T$. Then, the corresponding kinematical equations become

$$\vec{V_i} = \vec{V_a} + \vec{W} = \begin{bmatrix} \dot{x} \\ \dot{y} \\ \dot{z} \end{bmatrix} = V_a \begin{bmatrix} cos(\psi)cos(\gamma) \\ cos(\gamma)sin(\psi) \\ -sin(\gamma) \end{bmatrix} + \begin{bmatrix} u \\ v \\ w \end{bmatrix} \tag{3}$$

The equations of motion of the sailplane are obtained by applying Newton's second law

$$m\left(\dot{V_a} + \dot{u}cos(\psi)cos(\gamma) + \dot{v}cos(\gamma)sin(\psi) - \dot{w}sin(\gamma)\right) = -D + mgsin(\gamma)$$
$$m\left(\dot{\gamma}V_a - \dot{u}cos(\psi)sin(\gamma) - \dot{v}sin(\psi)sin(\gamma) - \dot{w}\right) = Lcos(\varphi) + mgcos(\gamma) \tag{4}$$
$$m\left(\dot{\psi}V_acos(\gamma) - \dot{u}sin(\psi) + \dot{v}cos(\psi)\right) = Lsin(\varphi)$$

Here, the inertial wind rates are:

$$\dot{u} = \partial u/\partial t + \partial u/\partial x\dot{x} + \partial u/\partial y\dot{y} + \partial u/\partial z\dot{z}$$
$$\dot{v} = \partial v/\partial t + \partial v/\partial x\dot{x} + \partial v/\partial y\dot{y} + \partial v/\partial z\dot{z} \tag{5}$$
$$\dot{w} = \partial w/\partial t + \partial w/\partial x\dot{x} + \partial w/\partial y\dot{y} + \partial w/\partial z\dot{z}$$

Combining Equations (4) and (5), the governing system of six first order differential equations can be written in the vector form:

$$\dot{\vec{Y}} = f\left(\vec{Y}, \vec{u}\right) \tag{6}$$

where $\vec{Y} = [x, y, z, V_a, \gamma, \psi]^T$ and $\vec{u} = [C_L, \varphi]$ are the state vector and control vector, respectively. Time dependent control parameters $C_L$ and $\varphi$ affect the flight trajectory, via either lift magnitude or heading. They are sufficient to control climb, descent, and turns of the three degrees of freedom model of the sailplane.

Performance of the sailplane is characterized by total mechanical energy $E = \Pi + K$, as the sum of the potential energy, $\Pi = -mgz$, and the kinetic energy $K = 0.5 \, mV_i^2$. Based on the developed dynamic model and specific mission requirements, sailplane flight patterns, using elements of gliding, and static and dynamic soaring maneuvers, can be optimally planned.

## 5. Soaring Methods

In this chapter we present two soaring methods for sailplanes. Conventional static soaring is discussed using vertical updrafts. Then, dynamic soaring is introduced with its reliance on non-uniform horizontal wind profiles. Finally, the latter is analyzed under wind conditions expected on Mars using numerical optimization and the sailplane's governing equations.

### 5.1. Static Soaring

Static soaring depends on the availability of vertical winds, including thermal and obstacle-induced slope soaring (Figure 4). Thermal soaring becomes viable in the presence of thermal updrafts, which are the result of uneven heating or cooling of the atmosphere (convection). They are known to exist on Mars, based on measurements collected during spacecraft entries [30], the observation of winds and dust devils [31], as well as atmospheric simulations [32,33]. Additionally, buoyancy-driven circulations over topography can induce daytime anabatic (upslope) winds [34,35] with vertical components that can reach several m/s.

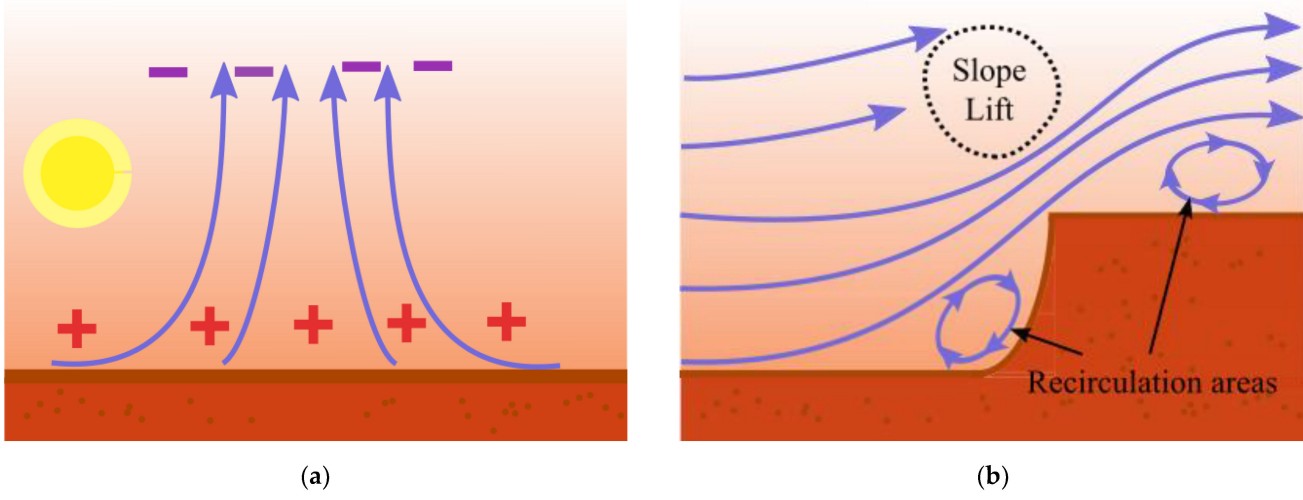

(**a**)                                    (**b**)

**Figure 4.** Atmospheric updrafts for static soaring: (**a**) thermals due to ground heating; (**b**) slope updrafts near cliff.

In the absence of naturally occurring thermal activity, slope soaring may be possible when prevailing winds are accelerated and diverted around geological features, such as

cliffs, steep slopes and ridges, inducing local updrafts, and recirculation areas (Figure 4b). The latter are generally undesirable for soaring, but under some conditions a favorable shear layer can exist prior to a recirculation vortex, which can be exploited, as is discussed later.

A sailplane can sustain flight when navigating in an updraft equal to its minimum sink velocity (6.28 m/s) and will gain potential energy if the updraft velocity exceeds the sinking speed. Practically, updrafts are exploited to gain altitude until the vertical winds weaken near the top, then the sailplane glides towards another updraft. Such patterns could help the sailplane reach distant science targets by alternating between thermal soaring, slope soaring, and gliding. Results from numerical modeling, using the Large Eddy Simulation (LES) method and mesoscale models, suggest that strong daytime convective updrafts exist in the PBL, extending to an altitude of up to 9 km [32]. For example, modeling work by [36] reports mean thermal velocities in the order of 5 m/s up to 5 km in altitude, while extrema are reported to reach 17 m/s [37], to 20 m/s [32]. By providing altitude gain, or, at the very least, by reducing the sinking velocity of the aircraft, the thermals can significantly extend the gliding range of the sailplane.

Note that from a mass-conservation standpoint, such strong updrafts are necessarily associated with return circulations (downdrafts). Daytime convective cells within the Martian PBL are thought to be of polygonal shape (Bénard cells), with vertical velocities reaching up to 8 m/s in the narrow updraft corridors that form the edges of the polygon, and boarder downdrafts existing at the center of the cells [36,38]. From a navigation (and instrumentation) standpoint, the sailplanes must be able to autonomously identify areas of strong thermal updrafts and use those corridors to navigate across flat areas when no obstacle-induced updrafts or slope flows are available; thus, extending range and flight time.

### 5.2. Dynamic Soaring

When the vertical velocity is zero, the sailplanes may engage in dynamic soaring maneuvers that solely require vertical gradients in horizontal winds, where the wind speed increases with altitude. They naturally occur in the planetary boundary layers over flat terrain (Figure 5a) and in shear layers over the leeward side of ridges or canyons (Figure 5b). Typically, shear layers are characterized by shallow and strong gradients, while winds gradients in the atmospheric boundary layer are weaker but span over a greater altitude range ("law of the wall" type of profile).

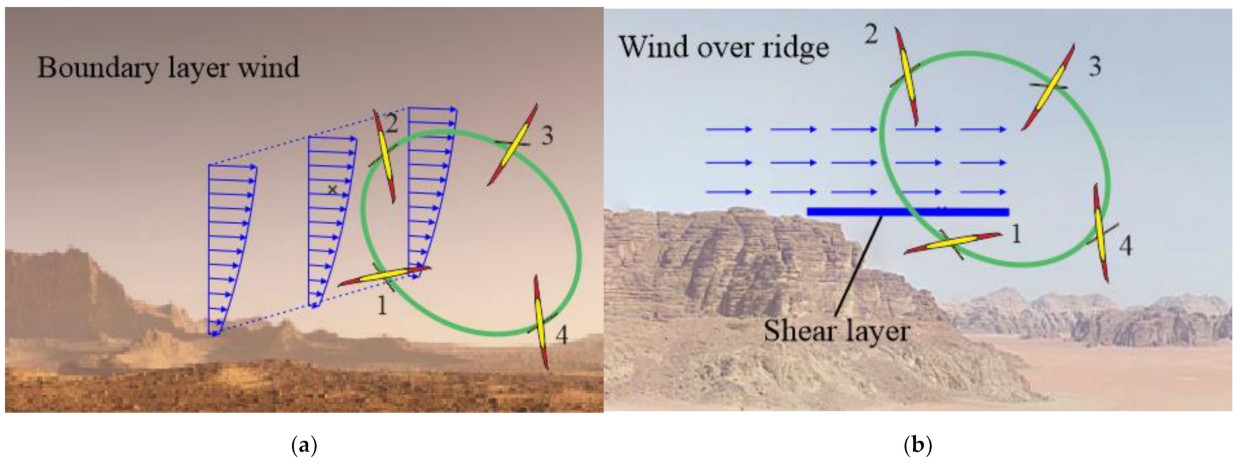

<div align="center">(<b>a</b>)                                                 (<b>b</b>)</div>

**Figure 5.** Dynamic soaring: (**a**) in atmospheric boundary layer (1—windward climb, 2—high-altitude turn, 3—leeward descent, and 4—low altitude turns); (**b**) in a shear layer developed over leeward region.

Dynamic soaring maneuvers comprise climb into non-uniform wind, high-g turn and descent. The sailplane flies into (or away) from winds with a vertical gradient and, by doing so, increases kinetic energy by harvesting the energy from the atmosphere. During this

maneuver, the average flight altitude can be maintained or gained providing continuous flight, as long as the minimal requirement for soaring conditions are met. Various flight patterns of dynamic soaring can be realized including loitering (Figure 5a,b), scanning (as in Figure 5a,b with small horizontal offset between passes) and advancing/traveling (Figure 1).

For a sailplane design (mass, aerodynamic performance, and sinking rate are fixed) the feasibility of dynamic soaring flight depends on existing atmospheric conditions. Numerical simulations were conducted using MRAMS [24] at Jezero crater, (Mars 2020 Perseverance rover landing site) during Northern spring, and at Melas Chasma in Valles Marineris during Northern summer. We used a set of 6 nested grids providing a horizontal resolution of ~1 km. Figure 6 shows the wind magnitude and vertical gradient for the Jezero crater site over one day. At this location, nighttime high wind gradients exist in the atmospheric boundary layer up to 1 km. Local horizontal wind profiles $u(z)$ and $v(z)$ were extracted from the simulations at local times $T_1 = 18:25$, $T_2 = 19:00$, and $T_3 = 20:00$ and shown in Figure 7. The flight altitude above the ground is defined as $H = -z$. Numerical characteristics of wind profiles are summarized in Table 2 for three altitudes: $H_0 = 0$, $H_2 = 170$ m, and $H_2 = 522.8$ m. This data shows relatively large winds and changes in boundary layer gradient.

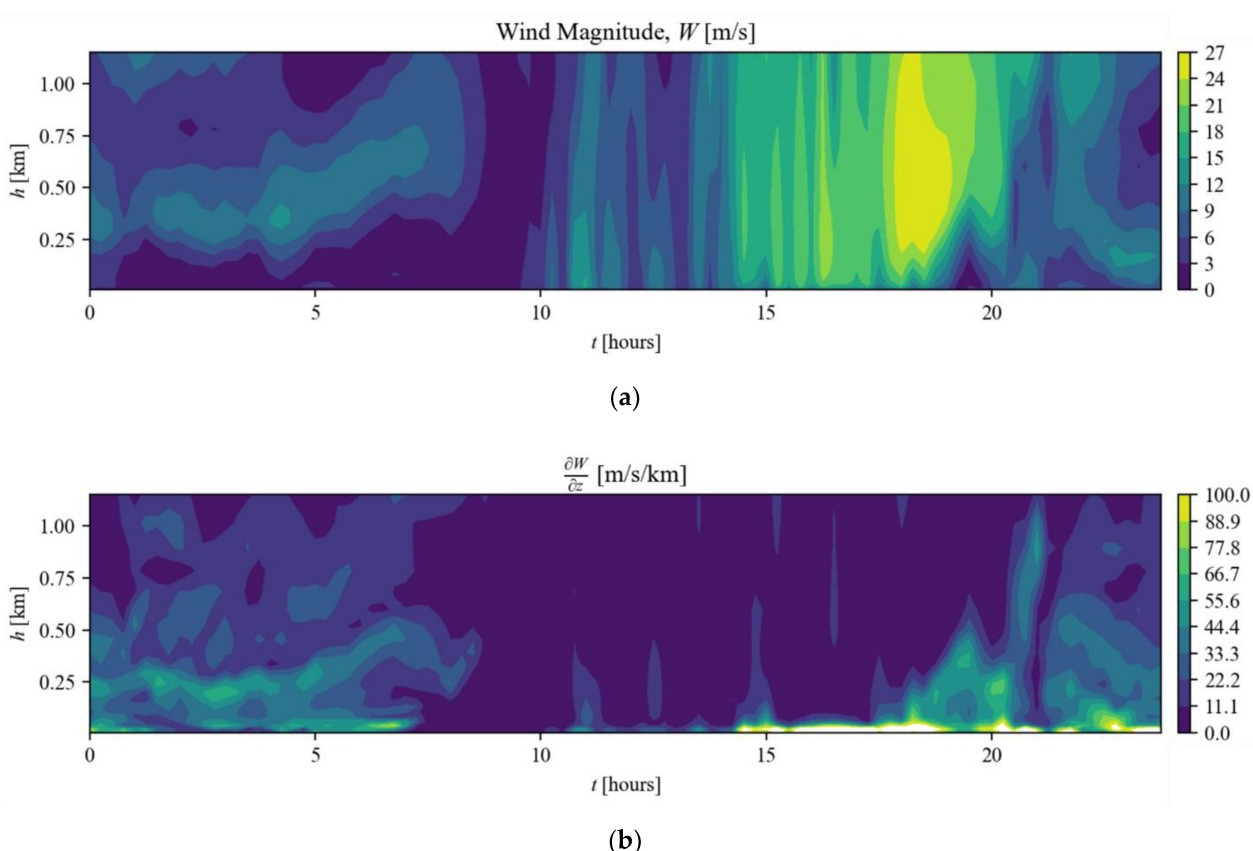

**Figure 6.** (**a**) Horizontal wind magnitude; (**b**) Vertical gradient of horizontal wind as a function of the local time and altitude from a MRAMS simulation at Jezero crater.

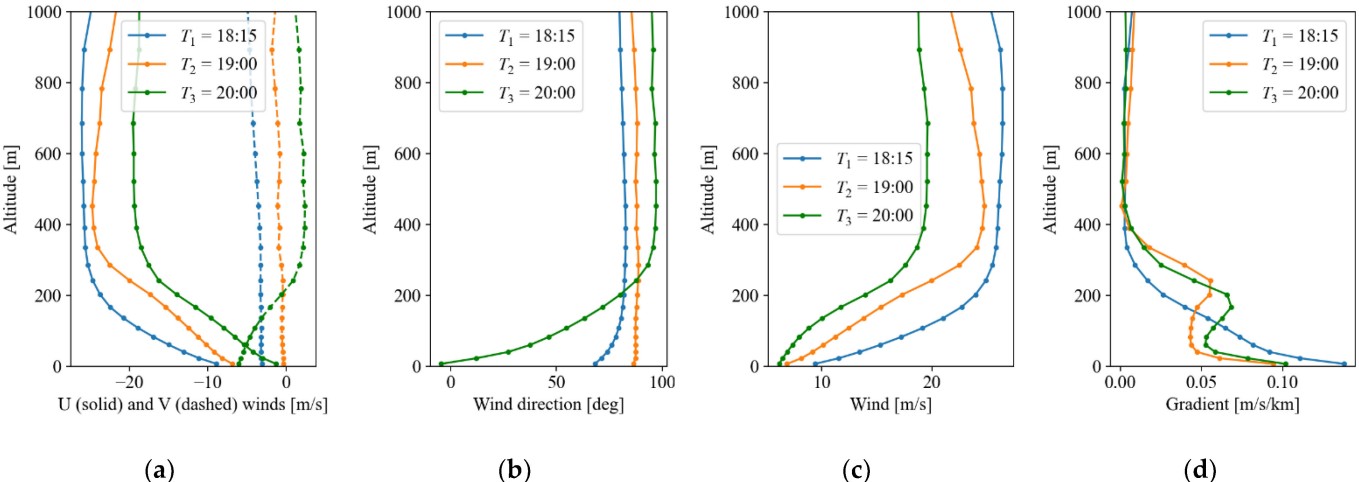

**Figure 7.** Wind magnitude profiles for selected time extracts from MRAMS data at the Jezero crater site. With (**a**) the west (U) and north (V) components of wind, (**b**) the wind direction, (**c**) the wind magnitude, (**d**) the vertical gradient in horizontal wind for each time.

**Table 2.** Characteristics of wind profiles.

| Time Instant | $T_1$ = 18:15 | $T_2$ = 19:00 | $T_3$ = 20:00 |
|---|---|---|---|
| $W_{H_0}$, [m/s] | 5.8 | 5.4 | 4.0 |
| $W_{H_1}$, [m/s] | 20.1 | 13.9 | 10.0 |
| $W_{H_2}$, [m/s] | 26.0 | 24.7 | 19.5 |
| $(dW/dz)_{H_0}$, [sec$^{-1}$] | 0.081 | 0.062 | 0.0590 |
| $(dW/dz)_{H_1}$, [sec$^{-1}$] | 0.054 | 0.044 | 0.062 |
| $(dW/dz)_{H_2}$, [sec$^{-1}$] | 0.0027 | 0.0005 | 0.0027 |

Consider the energy balance of the sailplane during a dynamic soaring cycle. The climb segment of the maneuver ends when the airspeed, speed $V_a$, approaches the stall speed, $V_{stall}$, or when the wind gradient drops below the minimum wind gradient value that is required for dynamic soaring. In the latter case, the kinetic energy of the sailplane, relative to the moving air defined as $K_a = 0.5mV_a^2$, starts to decrease, $dK_a/dt < 0$. The sailplane initiates a downwind turn and descends. At the end of the maneuver, the z-coordinate, and the pitch angle, $\gamma$, are approximately equal to that of the initial point. The dynamic soaring cycle is regarded as a successful one if the total energy of the sailplane increases during the cycle.

In order to find the flight trajectory that maximizes the sailplane energy at the end of the dynamic soaring cycle, a constrained optimization problem is formulated to maximize the rate of change of the kinetic energy relative to the moving air:

$$
\begin{aligned}
\max \quad & dK_a(t)/dt \\
s.t. \quad & |\gamma(t)| < \pi/2 \\
& |\varphi(t)| < \pi/2 \\
& V_{stall} < V_a(t) \leq V_{limit} \\
& C_{L\min} \leq C_L(t) < C_{L\max}
\end{aligned}
\tag{7}
$$

$V_{limit}$ corresponds to a Mach number of 0.8, which was selected intentionally as an operational limit to prevent excessive shock and the typical transonic region. The solution to the optimization problem gives a soaring cycle realized via control parameters $C_L(t)$ and $\varphi(t)$. The optimization of the non-linear differential equations was performed using the dynamic optimization algorithm with the Interior Point OPTimizer (IPOPT) solver for non-

linear problems, included in the GEKKO optimization library serving as an Application Programming Interface (API) in Python [39].

The overall energy efficiency of a maneuver is measured by the ratio of the energy change to the amount of energy at the beginning of the flight segment as

$$\Delta E(t) = (E(t) - E_0)/E_0 \tag{8}$$

Here $E(t)$ is total energy (potential plus kinetic energies) measured in the inertial frame of reference and $E_0$ is the energy at the start of the segment.

### 5.2.1. Numerical Optimization of Dynamic Soaring Segments

The subsequent numerical simulations characterize the energy efficiency of different segments of the dynamic soaring cycle. Given a simplified two-dimensional wind field $\vec{W}(z) = u\vec{x}$, two simulations were conducted. Firstly, we considered a reference 2-dimensional trajectory with a simple climb. Second, we considered a 3-dimensional trajectory with climbing and diving turns.

The analysis was performed using wind data presented in Figure 7. The wind profile is specified by the equation $\vec{W}(z) = u\vec{x}$, implying that the climb and dive maneuvers are held in the x-z plane. Therefore, simulations were conducted using a two-dimensional model by eliminating the roll controls: $\varphi = 0$, and $\psi = 0$; thus considering a subset of the three degrees of freedom state vector $\vec{Y}_2 = [x, z, V_a, \gamma]^T$. It gives the governing system of equations in the form

$$\dot{\vec{Y}_2} = f\left(\vec{Y}_2, C_L\right) \tag{9}$$

where $C_L(t)$ is a lift coefficient used as a control parameter, simplifying Equation (7) to a problem with four-time dependent states and a single time dependent control variable.

Flight trajectories and energy changes are shown in Figure 8a,b, respectively. Overall, climb and dive trajectories (Figure 8a) are similar with a slightly steeper climb profile. There are no constraints on final states and the state vector is $\vec{Y}_2(t_{cs}) = [0, -10, 80, 0^\circ]^T$ at the starting point of the climb the. The sailplane is at x = 0 and $H = -z = 10$ m travelling at 80 m/s in the negative x-direction. The sailplane gains the altitude of 283 m, while its airspeed reduces to 74.5 m/s. The total energy decreases by 14.7% relative to the start point of the maneuver. The dive starts at the same altitude and airspeed that the sailplane reached at the end of the climb. The dive ends at the altitude $H = 10$ m with an airspeed of 94.8 m/s, and total energy is reduced by 13.5%. Overall, these two segments are energy inefficient.

Secondly, we consider the case where roll control is nonzero, meaning the solution is not constrained to the x-z plane and the complete numerical model described by Equation (6) is solved separately for the two otherwise connected segments: climbing and diving turns.

At the beginning of the climbing turn, the state vector is identical to the previous zero roll case with $\vec{Y}(t_{cs}) = [0, 0, -10, 80, 0, 180^\circ]^T$. The climb ends at 439 m and $\psi = 90^\circ$ with an airspeed of 63.6 m/s. These are used as initial conditions in simulations of the diving turn. The airspeed at the end of the diving turn is 93 m/s at the altitude $H = 10$ m.

The trajectory of the climbing turn is steeper (Figure 8a). Figure 8b reveals that the energy graph is concave up and concave down for climbing and diving turns, respectively. At the end of the climbing turn, the relative energy, $\Delta E_{end} = 18\%$, is greater than that of the diving turn, $\Delta E_{end} = 8\%$. Note that both turning maneuvers gained a significant amount of energy compared to the in-plane dive and climb described previously.

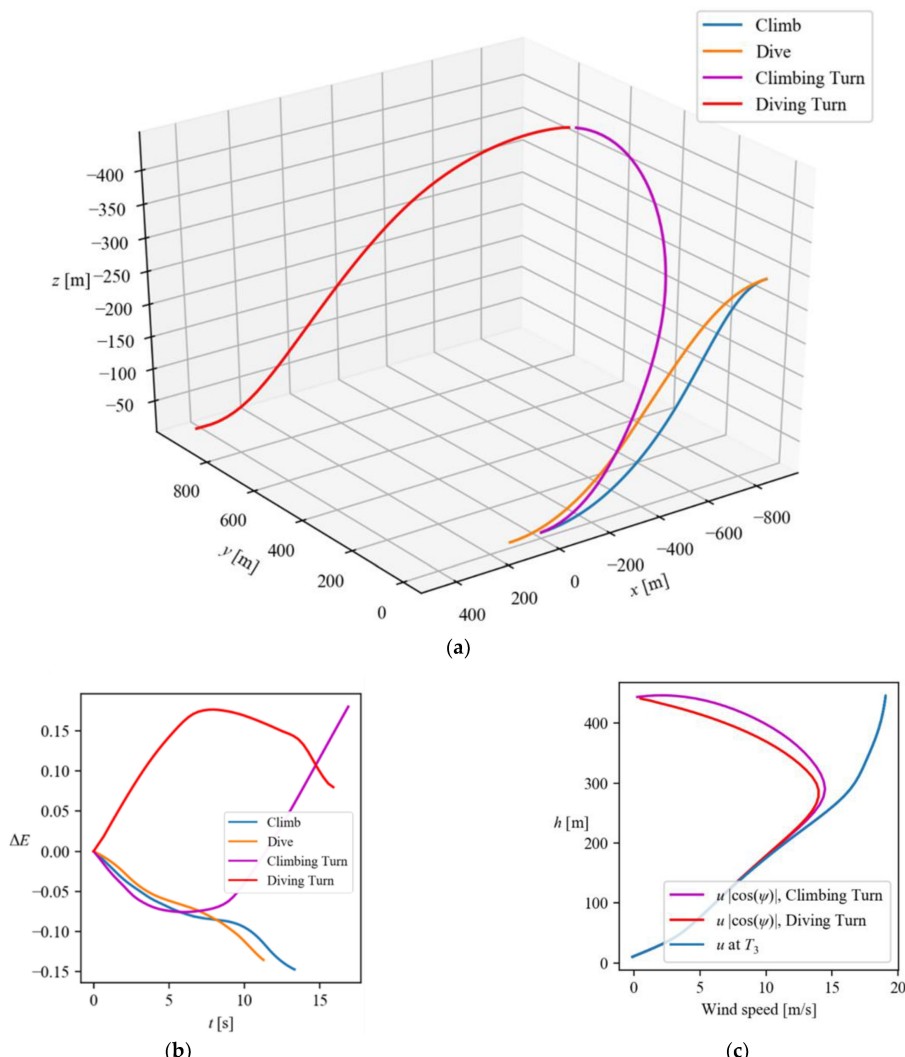

**Figure 8.** Results of simulations of climb, dive and turning segments of dynamic soaring, subject to simulated unidirectional wind at time $T_3$. (**a**) flight trajectories; (**b**) time change of total energy; (**c**) wind projections.

Since the aerodynamic side forces and the yawing moment are not included in the present dynamics model, the effect of the wind projection from the side direction was effectively ignored. Therefore, the sailplane is exposed to the projection of the wind vector $\vec{W} = u\,\vec{x}$ on the unit vector $\vec{e} = cos(\psi)\,\vec{x} + sin(\psi)\,\vec{y}$. The magnitude of this component of the wind vector, $uc°s(\psi)$, during turning maneuvers is shown in Figure 8c, along with the nominal wind $\vec{W} = u\,\vec{x}$ taken from Figure 7. The former is found to decrease as the altitude increases. At the end of the climb and the start of the dive the wind is perpendicular to the flight path. It is seen that wind gradient increases significantly during the first and last 7 s of diving and climbing turns, respectively, which corresponds to altitudes of ~300 to 400 m in Figure 8c. The wind gradient "seen" by the sailplane results in an increase of the total mechanical energy of the sailplane (Figure 8b). This reveals that optimizing for the change in kinetic energy relative to the air (7) drives the sailplane into turning trajectories with a higher effective wind gradient.

### 5.2.2. Numerical Optimization of Advancing Dynamic Soaring Cycles

Results for a complete dynamic soaring cycle were obtained using three-dimensional dynamics model Equation (6) and the optimization objective described in Equation (7),

with the final constraints $z\left(t_f\right) = z(0)$, $\psi\left(t_f\right) = \psi(0)$, $\gamma\left(t_f\right) = 0$. The wind velocity was presented in the form: $\vec{W} = u\vec{x} + v\vec{y}$, where data for $u = u(z)$ and $v = v(z)$ are given in Figure 7 for the three wind profiles under consideration at local times $T_1 = 18{:}15$, $T_2 = 19{:}00$, and $T_3 = 20{:}00$. In all cases the initial conditions are identical and equal to the state vector $\vec{Y}(t_{cs}) = \left[0, 0, -10, 80, 0, 180^\circ\right]^T$.

Figure 9a shows the resulting flight trajectories of the sailplane for the three local times. The sailplane travels holding a flight direction at approximately $60^\circ$ to the prevailing wind direction seen in Figure 9a. Table 3 summarizes performance characteristics of each flight: the distance travelled, $S$, the maximum altitude reached, $H_{max}$, and the energy change at the end point, $\Delta E_{end}$.

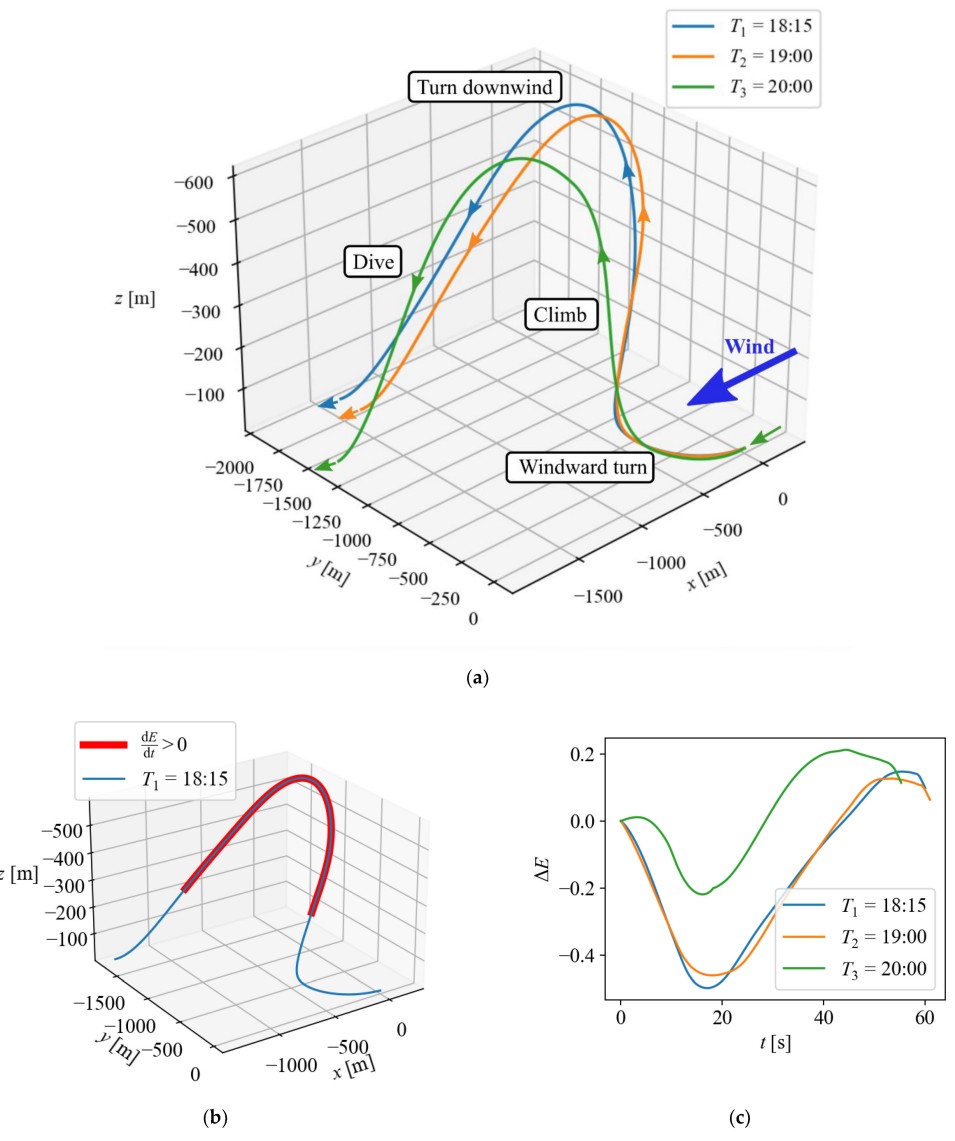

**Figure 9.** Dynamic soaring cycles. (**a**) sailplane trajectories for three Mars times; (**b**) a single cycle with positive $dE/dt$ markers; (**c**) energy time history for each cycle.

**Table 3.** Characteristics of dynamic soaring cycles.

| Time Instant | $T_1 = 18{:}15$ | $T_2 = 19{:}00$ | $T_3 = 20{:}00$ |
|---|---|---|---|
| $H_{max}$, [m] | 610 | 587 | 520 |
| $S$, [m] | 2400 | 2200 | 2300 |
| $E_{rend}$ | 0.098 | 0.063 | 0.113 |

These solutions to the optimization problem described in Equation (7) first drives the sailplane to use its energy for turning windward, as evident in Figure 9b,c. Next, the sailplane performs a turning climb and its energy increases, a maneuver proven favorable in Figure 8. After reaching the maximum altitude $H_{max}$ = 520 to 610 m, the sailplane turns downwind. Note that at $H_2$ = 523 m, the wind gradients, $(dW/dz)_{H_2}$, for all three times, is close to zero (Table 2). Downwind diving begins as the airspeed is close to stall, it then recovers airspeed as potential energy converts into kinetic energy. Finally, the cycle ends at approximately the same altitude and attitude as at the starting point. The energy at the end point, $\Delta E_{end}$, increases in the range from ~7% to 11% for the three studied times (Table 3). Thus, energy was harvested during these dynamic soaring cycles, while the aircraft advanced along the travel path by 2200–2400 m.

When testing for different wind profiles (different times of day), we verified that changes in total energy correlate well with the strength of the wind, and that dynamic soaring is a viable maneuver for unpowered flight for a range of wind conditions.

While wind magnitudes and gradients at the time $T_1$ are larger (except the wind gradient $(dW/dz)_{H_1}$ at $H_1$ = 170 m) compared to the two other times, corresponding values of distance travelled, maximum altitude and relative energy at that time are greater (Table 3).

We observed that a wind gradient combined with significant wind direction changing over altitude (Figure 7, time $T_3$) presents an advantage over more steady conditions (times $T_1$ and $T_2$). Such changes in wind direction cause the windward turn to become part of the energy harvesting phases of flight (Figure 9c.), reducing the overall amplitude of energy change over a dynamic soaring cycle and increasing the overall energy available to the sailplane.

## 6. Static Soaring and Flying in Canyons

We propose several flight patterns to provide the close oblique views necessary for the geological survey of canyons and their stratigraphy. They consist of the following segments: gliding, diving, pullout of the dive, climbing, static.

Gliding, the simplest flight mode usable for reconnaissance missions, is described by the minimum sink rate of the sailplanes. It shows the ability of the sailplane to perform long-range flights without assistance from atmospheric winds. For the current sailplane design, the minimum sink rate was found to be 6.28 m/s determined at the maximum lift-to-drag ratio $C_L/C_D$ = 12.04, angle of attack $\alpha = 5.06°$, and airspeed $V_a$ = 62.65 m/s. Therefore, from the altitude of 6 km above the aeroid, the craft can glide 12.04 × 6~72 km and potentially more if the glide path brings the sailplanes to the areas below zero elevation. For example, assuming a science target at the bottom of Melas Chasma, in Valles Marineris (elevation approximately −3 km below the aeroid), the altitude difference is 9 km, which translates to an achievable range and flight time of about 100 km and 20 min, respectively. Ultimately, this means that the requirement for the EDL system (e.g., landing ellipse) is relaxed by as much as 200 km, which ultimately simplifies the design of the entry system.

For the following maneuvers in the list, Figure 10 illustrates a use case where a sequence of such maneuvers is applied to the exploration of canyon walls. The flight begins at 4 km initial altitude above the rim of a canyon. The sailplane dives deep into the canyon along its walls. After reaching the lowest point, it pulls out to the maximum recovered altitude. Then, static (slope flows) and/or dynamic soaring maneuvers are performed to regain the altitude and energy of the sailplane lost during the dive-pullout segment of the flight. The cycle can then be repeated, allowing for sustained sailplane flight.

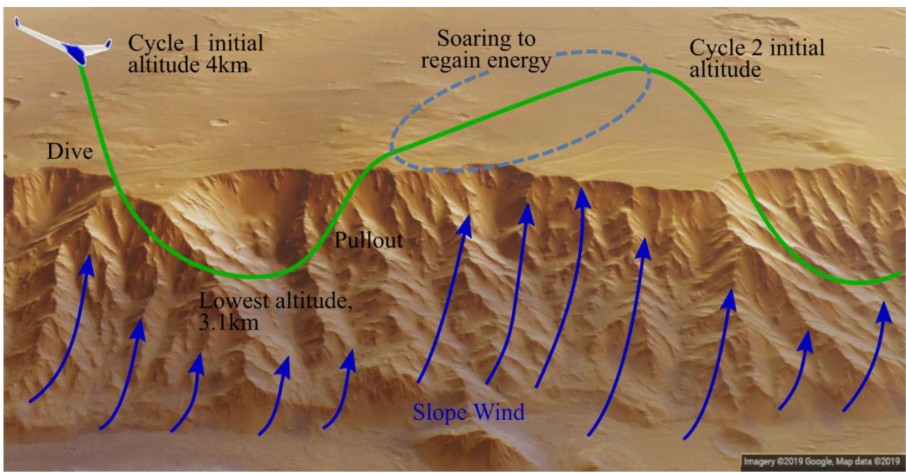

**Figure 10.** Canyon exploration concept using dive-pullout maneuver followed by static or dynamic soaring.

The origin of the frame of reference is placed at the canyon's ridge with the z-axis pointing downwards, thus the flight height is $H = -z$. The sailplane dive-pullout flight trajectory was analyzed by splitting the maneuver into dive and climb segments. The dive starts at the time $t_{ds}$, ends at $t_{de}$, and the climb lasts from $t_{cs} = t_{de}$ to $t_{ce}$. The state and control vectors at the end of a dive determine the starting conditions for the next climb. A zero-pitch angle is imposed at the start and the end of the dive-pullout maneuver, i.e., $\gamma(t_{ds}) = \gamma(t_{ce}) = 0$.

The dynamics system Equation (9) is utilized for a two-dimensional case describing a dive-pullout maneuver. No wind is assumed in the simulations of dive-pullout flights, thus $\vec{V_i} = \vec{V_a}$.

The reachable depth within the canyon is constrained by the sailplane's ability to regain potential energy from the kinetic energy accumulated during the dive and finishing the climb at a height greater, or equal to, the canyon's ridge ($H = 0$. Then, for the climb segment, the objective is to maximize a recoverable height $\Delta H = z(t_{cs}) - z(t_{ce})$ at the end of the maneuver

$$
\begin{aligned}
\max \quad & \Delta H \\
s.t. \quad & |\gamma(t)| \leq \gamma_c \\
& V_{stall} \leq V_a(t) \leq V_{limit} \\
& C_{L\min} \leq C_L(t) \leq C_{L\max}
\end{aligned}
\tag{10}
$$

As for the task of finding a dive trajectory, the solution is equivalent to maximizing the airspeed at the end point, $t_{de}$, and solved by expressing an optimization problem as

$$
\begin{aligned}
\max \quad & V_a(t_{de}) \\
s.t. \quad & |\gamma(t)| \leq \gamma_d \\
& z(t) \leq z_{max} \\
& V_{stall} \leq V_a(t) \leq V_{limit} \\
& C_{L\min} \leq C_L(t) \leq C_{L\max}
\end{aligned}
\tag{11}
$$

Overall, both problems are joined as $z(t_{cs}) = z(t_{de})$ and both objectives affect this maximum reachable depth within the canyon.

Studies of the effect of the dive steepness were conducted for three values of the maximum achievable pitch angle during the dive: $\gamma_d = 30°, 60°, 80°$. At the beginning of a dive, the state vector $\vec{Y_2}(t_{ds})$ is $[x, z, V_a, \gamma]^T = [0, -4,000, 55, 0]^T$ ($H = 4$ km above the canyon and 55 m/s of airspeed). Obtained trajectories are presented in Figure 11a. As expected, the sailplane reaches maximum speed and kinetic energy near the lowest point of flight trajectory, $z_{max}$, and then pulls out of a dive. In the presented examples, the maximum reachable airspeed at the bottom of the dive, $V_{de} = 187.13$ m/s, was achieved at the steepest

dive trajectory with the flight path angle of $\gamma_d = 80°$. This speed also defines the maximum achievable canyon depth under the given flight conditions via the recoverable $\Delta H$ when $V_{cs} = 187.13$ m/s.

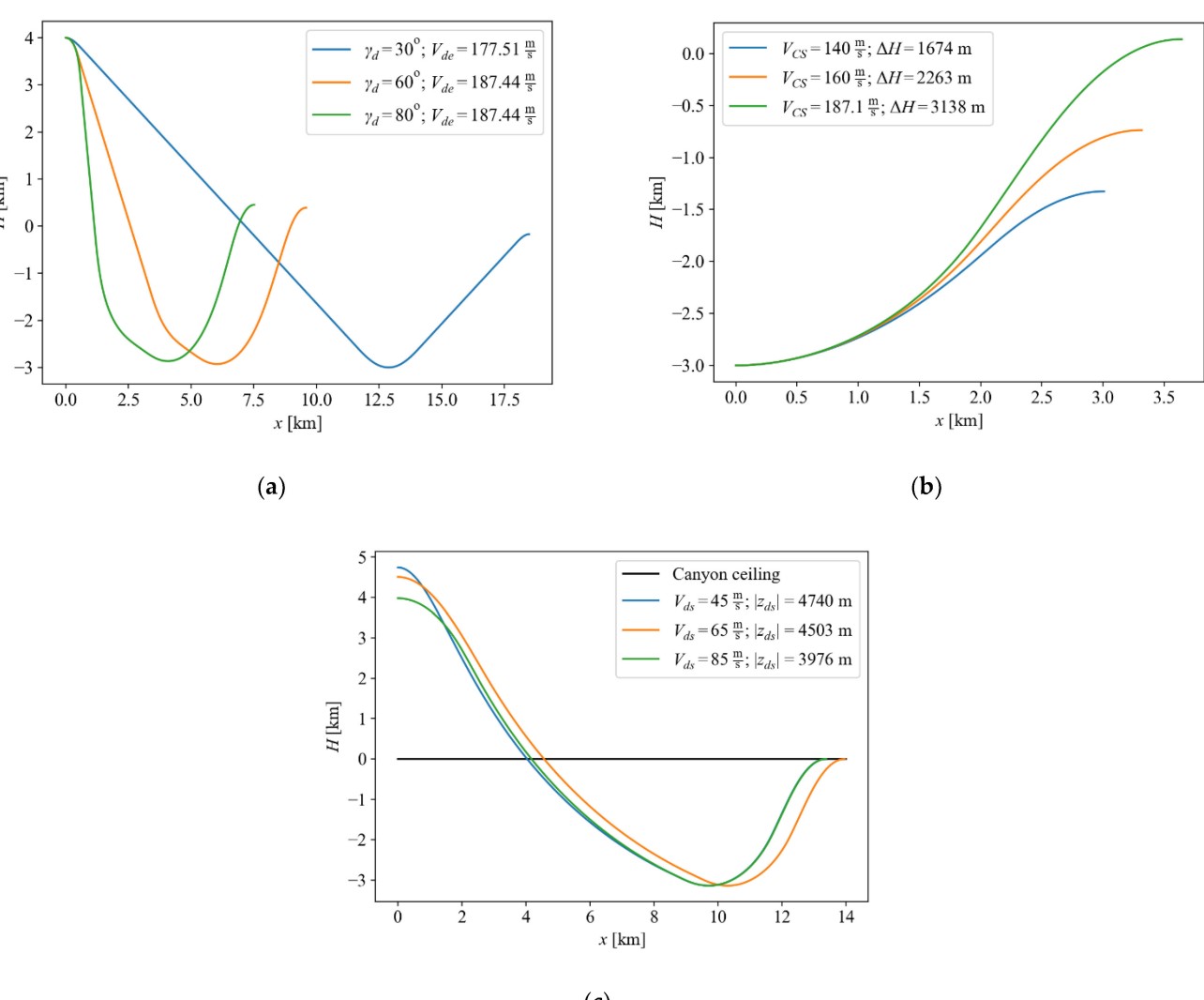

**Figure 11.** Sailplane trajectories for: (**a**) dive-pullout at three values of pitch angle, $\gamma_d$; (**b**) climb segment at three values of airspeed $V_{cs}$; (**c**) dive-pullout for three pairs of $V_{ds}$ and $z_{ds}$ at the beginning of a dive.

The climb segment was analyzed separately with the state vector at the start point $\vec{Y_2}(t_{cs}) = [0, 3,000, V_{cs}, 0]^T$ ($H = -3$ km, below the canyon ridge). Flight trajectories were obtained for three values of $V_{cs} = 140, 160, 187.1$ m/s. Numerical solutions presented in Figure 11b give the maximum recoverable height $\Delta H = 3138$ m. This corresponds to the maximal canyon depth that can be reached with the current sailplane design. Future improvements in the aerodynamic design of aircraft, or the use of slope winds to recover altitude, will be necessary to achieve lower diving depths for the exploration of canyons.

Figure 11c draws a comparison between trajectories of three pairs of initial velocity, $V_{ds}$ of 45, 65 and 85 m/s and altitude, $z_{ds}$. The trajectories were calculated for the following kinematic parameters: $\gamma_d = \gamma_c = 80°$, $z_{max} = 3000$ m. For a given $V_{ds}$, the altitude at the starting point, $z_{ds}$, was determined iteratively providing the same end speed $V_{de} = 187.13$ m/s for all three cases. Consequently, all three maneuvers generate the same airspeed and altitude at the end of the climb. When the initial velocity increases almost 2 times, the required initial altitude is lower by only 20% (Figure 11c). This result can be

explained by a large relative potential energy of the sailplane at the beginning of the flight dominating flight kinematics in this case.

When a dive-pullout maneuver is followed by static or/and dynamic soaring, multiple dive-pullouts along the canyon walls are possible in the Martian atmosphere, extending the traverse path over long distances or providing opportunity to multi passes after a turn-around maneuver is performed. Future research will attempt to characterize atmospheric flows in canyons, including wind studies in the context of static and dynamic soaring.

## 7. Sailplane Design and Deployment Techniques

Sailplanes packaged into CubeSats will occupy some of the available ballasts (about 190 kg on the Mars Science Laboratory-class mission). They will be deployed during the descent phase of atmospheric insertion, as the vehicle slows down to 100 m/s lateral velocity and achieves an altitude of 7 km, at which point the Mars Sailplane Aircraft (Figure 12) will separate from the EDL vehicle [40].

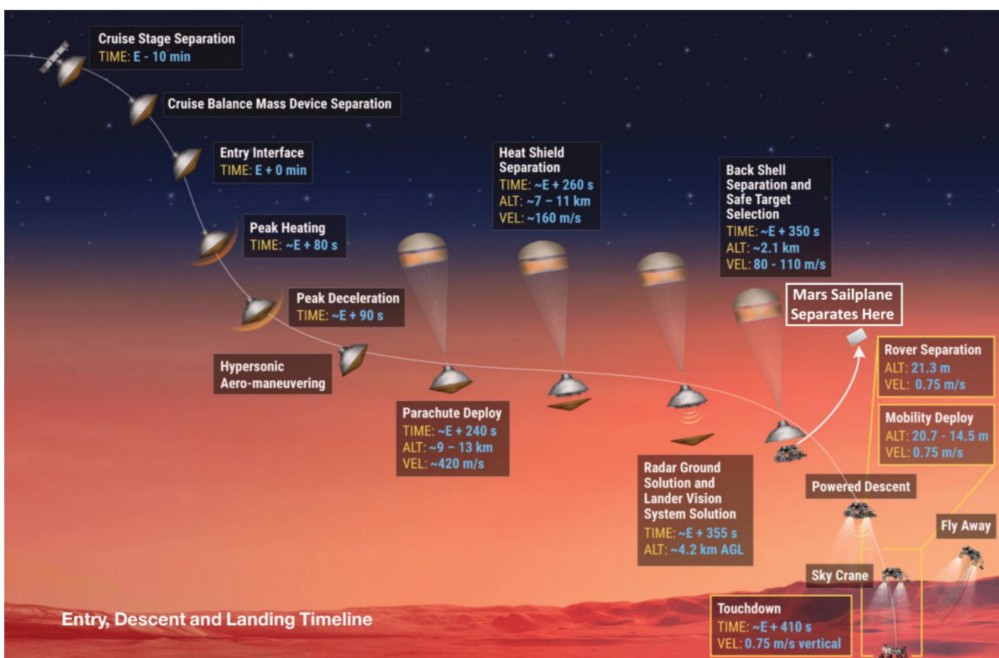

**Figure 12.** Entry, Descent and Landing Sequence for the Mars Science Laboratory/Mars 2020 Rover System.

The sailplane will undergo a 5–15 s deployment sequence, starting with a cold gas propulsion burn to achieve 1 km of separation distance with the insertion vehicle, reaching the initial gliding phase at 6 km altitude. In the proposed design, each sailplane is equipped with an MSSS ECAM-C50 reconnaissance camera [41], NASA JPL IRIS v2 X-band communications, a flight control system, and environmental instruments, totaling 5 kg.

Two designs for fast deployment of sailplanes are being considered. One is a roll-up inflatable wing design. The second is an "accordion" fold design of the inflatable wing. In the first design, the gas generation system from the lander airbag, used on the MER and Pathfinder missions, will be packaged into a 6U form-factor CubeSat attached to another 6U for all the remaining components of the aircraft (Figure 13). The gas generation system will produce non-toxic $N_2$ and pressurize the wing within 3 s [42]. Telescopic Astro-tube booms, developed by Oxford Space Systems [43], will expand to provide full structural support for the wing structure within ~10 s, followed by disposal of gas generator and boom deployer. Inflatable structures have been demonstrated during EDL on Mars Pathfinder and Mars MER Rovers [44,45].

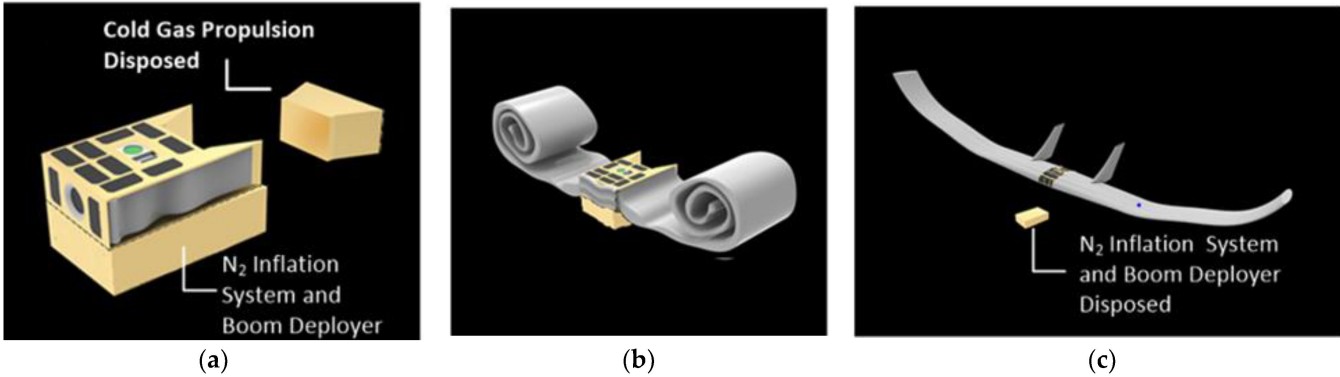

**Figure 13.** Mars sailplane deployment steps. (**a**) The cold-gas propulsion is used to achieve a separation distance; (**b**,**c**) Wing deployment begins.

The inflatable wing structure would contain a modular rib spar and a pair of extendable carbon-fiber Astro-tube booms [43]. The Astro-tube boom is shown to lift or hold a 300 g payload at its tip in Earth gravity. Each Astro-tube boom would expand 1.675 m to provide support to the wing.

A second design configuration has some promising advantages over the roll-up design. This can deploy faster and achieve a rigidized shape (Figure 14) in fewer seconds. This inflatable wing has a rigid support structure that starts off like an 'accordion', unfolding flat using a torsional spring, and rigidizes as it locks into place. As the inflatable system is proven to deploy in less than 3 s, we can expect a rigid wing to be deployed within 3–5 s. This is concurrently followed by nitrogen filling the inflatable.

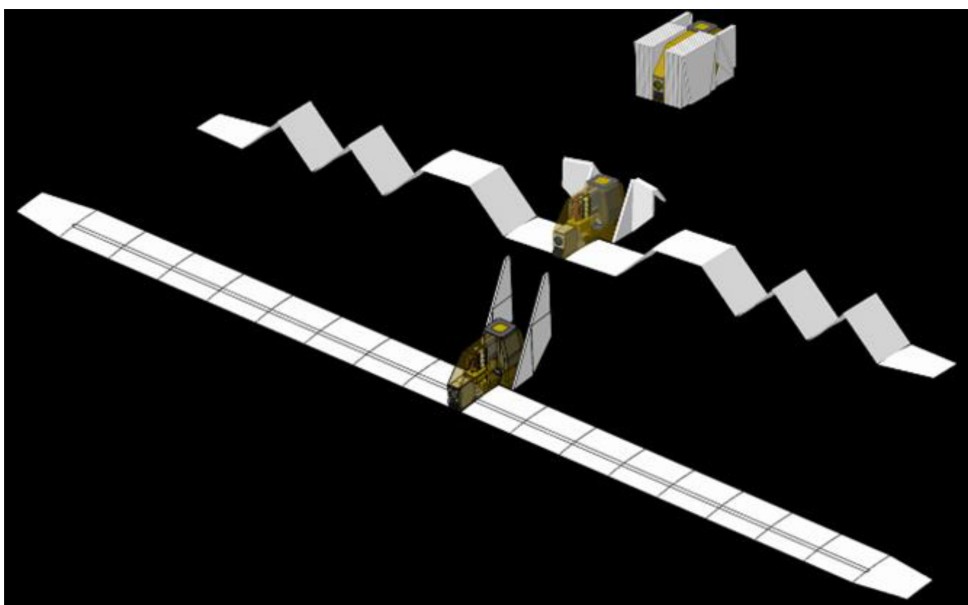

**Figure 14.** Inflatable wing design using an accordion fold system.

A credible alternative to direct deployment of the sailplane during EDL is to first deploy a balloon [46–48] or a blimp. These aircraft would, in turn, carry the sailplane deployment package and deploy the package at a more leisurely pace in the order of hours. Balloon or blimp assisted deployment could significantly de-risk the sailplane deployment sequence. As there is a large buoyant force available to keep the deploying Mars sailplane afloat in the atmosphere, there is less concern for achieving lift and stability control from the sailplane itself. Once the sailplane has deployed, it could be commanded to separate from the balloon or blimp at the optimal time and location to perform diving and dynamic

soaring maneuvers for science reconnaissance. After a few minutes to an hour of flight, the sailplane returns to dock with the balloon or blimp. A second option could be to deploy a blimp that can carry multiple sailplanes (Figure 15).

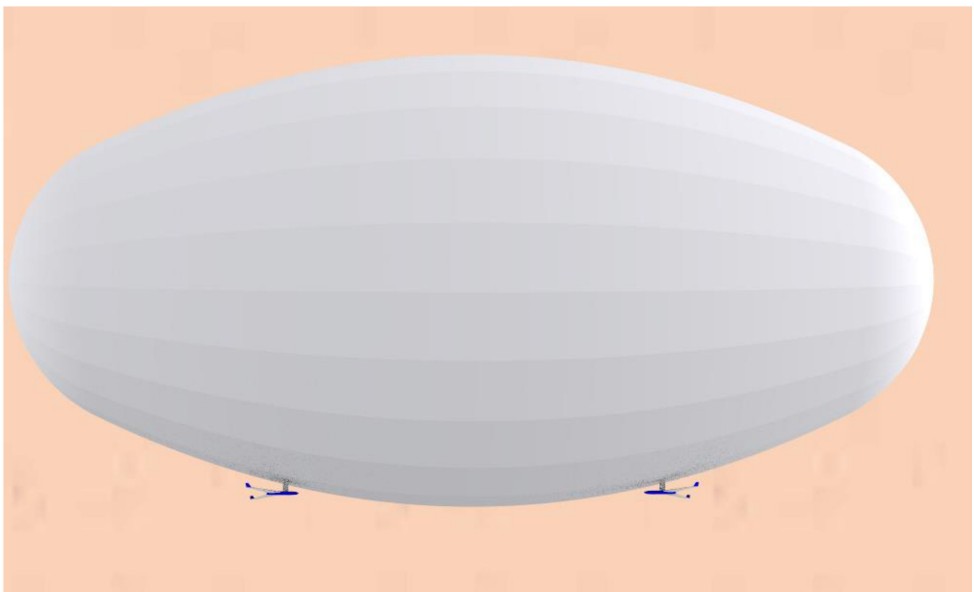

**Figure 15.** Pair of Mars Sailplanes mounted to a blimp (drawn to scale).

The balloon or blimp could be either a lighter-than-air filled super pressure balloon or a solar heated Montgolfier. Super pressure balloons have the advantage of flying multiple days but have an added logistical challenge of transporting the lighter-than-air gas and added inflation complexity. Alternatively, solar hot air balloons have been proposed [46,47] that heat the internal gas via direct solar radiation and can fly as long as the sun is visible. These balloons can also be deployed in the air and inflate via ram-inflation [49].

One or more sailplanes mounted to a balloon would have the advantage of increased direction and altitude potential. Such a system has redundancies that could make long duration voyages, lasting weeks to months, credible. It can enable transport from the landing region (primarily located near the equator) to regions of interest, such as Valles Marineris, Melas Chasma, Cydonia, Olympus Mons and the Southern Highlands.

This two-sailplane system also provides the ability to conduct coupled missions (Figure 1) without multiple atmospheric insertions from orbit. Redundancy for primary objectives also arise since, if one sailplane is lost, the mission continues. Secondly, if there is any difficulty in reaching the blimp, the blimp in turn could be commanded to move towards the sailplane to enable a rendezvous maneuver, or ground recovery. Furthermore, these rendezvous attempts can be carried out multiple times to ensure success. It should be noted that the balloon alone could not achieve what is possible with the balloon-sailplane hybrid system. The sailplane is smaller and has more maneuverability to dive, get around edges of cliffs, canyons, and deep valleys, and get viewpoints that would not be possible with the blimp alone.

## 8. Conclusions

The present work has expanded the range of possible methods to sustain flight for the exploration of Mars using sailplanes that harvest energy available in atmospheric flows, considering problems associated with low atmospheric density and low Reynolds numbers. Such platforms would be compact enough to be deployed as secondary payloads, and enable breakthrough mapping of large areas, including rugged terrains previously not accessible to rovers and landers. Equations of motion of the sailplane were derived, and optimal flight trajectories were found by formulating an optimization problem with

lift coefficient and roll angle as control parameter. We studied three types of maneuvers than can be used sequentially to keep the sailplane aloft: (1) static soaring in thermals or obstacle-induced updrafts, (2) dynamic soaring which exploits vertical gradients in horizontal winds, (3) dive-in and pull-out maneuvers along canyon ridges. Key results from the study are summarized as follows:

*Static soaring.* With the selected aerodynamic design, a sailplane with a baseline weight of 5 kg has a ~6.3 m/s sink velocity, implying updrafts in the order of 6.3 m/s are necessary for static soaring. Modeling results from both our own simulation and from the literature suggest such daytime vertical velocities are locally available on Mars near prominent orography.

*Dynamic soaring.* The energy efficiencies for different segments of the dynamic soaring cycle were analyzed. Turning maneuvers gained significantly more energy compared to in-plane dive and climb maneuvers. Numerical simulations revealed that the total energy graph is concave up and concave down for climbing and diving turns, respectively, and the increase in energy over one soaring cycle correlates well with the wind gradient. Predictions from a mesoscale Mars climate model were used to provide reference Martian winds, and it was demonstrated that nighttime wind gradients provided conditions suitable to dynamic soaring for a case study at Jezero crater.

Numerical results for complete dynamic soaring cycles were obtained for the travelling flight pattern, in which the sailplane holds the flight direction at approximately $60^\circ$ to the wind direction. Firstly, the optimal solution drives the sailplane to use its energy for turning windward and climbing. The sailplane reaches the maximum altitude at airspeed close to stall and turns downwind. The potential energy of the sailplane then converts into kinetic energy, recovering airspeed. The total sailplane energy at the end point of the soaring cycle increases by 7–11%, while the aircraft travelled 2200–2400 m forward.

*Canyon soaring.* The sailplane dive-pullout trajectories were analyzed flying various trajectories along a portion of the Valles Marineris. Numerical results demonstrated that Mars sailplanes can do close-to-wall flying passes over locations inaccessible by conventional landers and rovers, thus providing a unique, close-up oblique viewing of the canyons and their stratigraphy.

*Aerodynamic design.* Our baseline design uses a 1.8 m$^2$/3.35 m long sweptback flying wing with the S9033 airfoil. The thin airfoil was selected to provide high lift-to-drag ratio, moderate pitching moment coefficient, and the sweptback wing configuration provides additional lift, due to its blended-wing-body design.

*Mechanical design and deployment.* Two designs for fast deployment of sailplanes were proposed: a roll-up inflatable wing design and an "accordion" fold design for the inflatable wing. The sailplanes are packaged in a 12U form factor (two 20 × 30 × 10 cm units) with a 6U, 5 kg upper fuselage containing navigation camera, radio, flight control systems, and the wings and a 6U detachable lower bus, containing the wings' deployment and inflation systems as well as a propulsion unit to ensure initial separation/attitude correction from the main spacecraft when carried as a secondary payload.

Finally, various methods of the sailplane deployment in Mars's atmosphere were presented, including quick deployment during EDL and deployment using balloons/blimps. Balloons could significantly de-risk deployment challenges and transport the sailplanes long distances on extended missions to targeted regions, while exploiting maneuverability of the sailplane(s) for access into steep canyons, cliffs, and crater walls. It is the combination of sailplane and blimp/balloon that brings significant exploratory advantage, rather than either alone.

**Author Contributions:** Conceptualization, A.B., A.K., S.S. and J.T.; methodology, A.B., A.K., S.S. and J.T.; software, A.B.; validation, A.B.; formal analysis, A.B., S.S. and A.K.; investigation, A.B., A.K., S.S., J.T. and T.S.; resources, S.S. and J.T.; writing—original draft preparation, S.S.; writing—review and editing, typesetting, A.B. and H.K.; visualization, A.B. and T.S.; supervision, S.S., A.K. and J.T.; funding acquisition, S.S. and J.T. All authors have read and agreed to the published version of the manuscript.

**Funding:** This research was funded by the University of Arizona Micro Air Vehicles Laboratory. Tristan Schuler and Himangshu Kalita were supported in part by National Aeronautics and Space Administration, grant number 80NSSC19M0197.

**Institutional Review Board Statement:** Not applicable.

**Informed Consent Statement:** Not applicable.

**Data Availability Statement:** Not applicable.

**Acknowledgments:** The authors would like to acknowledge with pleasure the discussions with Alfred McEwen. In addition, the authors would like to acknowledge Aman Chandra for developing cad figures, Figures 14 and 15.

**Conflicts of Interest:** The authors declare no conflict of interest.

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
