# Peer review of "Mars Exploration Using Sailplanes"

_aerospace, doi:10.3390/aerospace9060306_

Round 1

Reviewer 1 Report

This manuscript “Mars Exploration Using Sailplanes”proposes a new preliminary design of sailplanes, used for planetary exploration of Mars. For the flight of the sailplanes on Mars, this paper mainly completes two innovative works: Firstly, the flight pattern is proposed, which exploits atmospheric wind gradients for dynamic soaring, and slope/thermal updrafts for static soaring. The derivation of the corresponding dynamic model optimal trajectory design and case simulation are completed based on MRAMS. Secondly, in order to achieve fast deployment of sailplanes, innovative methods of the sailplane deployment in Mars’s atmosphere are presented.

This is a well-written paper and clearly expounds the research results through rigorous theoretical derivation and many simulation examples. And the submission is worth of publication. The paper’s achievements are of considerable interest to the researchers in the related areas but needs some improvement before acceptance for publication. My detailed comments are as follows:

  1. The part about the sailplane deployment and unfolding in the abstract could be written more concisely.
  2. In the introduction, the references to previous research are insufficient. The research status and existing problems of Mars exploration missions are briefly described through only a few missions. And no background description on the design of sailplane deployment is provided.
  3. Some keywords are not the main focus of the article,such as “CubeSat”. Some keywords involve too broad fields, you can choose more specific and detailed keywords, such as “Mars” can be changed to “Mars exploration”.
  4. It is a bit redundant and unnecessary to divide the article into eight chapters. It is suggested to further optimize the structure of the article, so that the structure of the article is more reasonable, clear, and logical. For example, chapters “Static Soaring” and “Dynamic Soaring” can be written in two subsections of a same chapter, instead of being separate chapters.
  5. Chapter 4 “Numerical Simulations of Flight in Martian Atmosphere” only deduces the kinematic equations of the sailplane in the Martian atmosphere, and does not carry out any numerical simulation of the simulation example, so the naming of the chapter needs to be considered.
  6. This paper demonstrates a large number of simulation cases, and it is better to give a clearer description of each simulation example. For example, in Chapter 6, several simulation examples are slightly chaotic. It is recommended that the conditions, results, and analysis of the examples could be more clearly stated. And two cases need to be clearly separated.
  7. Many graphs and diagrams have unclear lines and text, I suggest changing the way the pictures are generated to get a higher definition picture to insert into the paper. For example, the axes and legend of Figure 8,9 are not clear; The text in Figure 1,3,5 is not clear, and the resolution of the picture is too low.

Reviewer 2 Report

Please find comments and suggestions as an attachment.

Round 2

Reviewer 1 Report

There are some details that need to be addressed.

1. The last sentence of the revised abstract has no semantic meaning and has repetition:

Original text:Various methods for sailplane deployment are considered, including fast deployment during Entry, Descent and Landing (EDL) of a Mars Science Laboratory-class (MSL) vehicle, deployment using blimps are presented.

2. Some units and symbols in the article are not standardized enough. Such as in “ Aerodynamic Design of Sailplane”,“g = 3.7278 kg m/s2” and “1.08⋅10-5 N s/m2”(page 4) should use superscripts, and in “7. Sailplane Design and Deployment Techniques”, “CO2 atmosphere” (page 18) should use subscript.

3. The information described in some tables in the article is not clear enough, especially the description of units. For example, the units in the "parameters" column in Table 1 could be expressed as [m] or [m/s] using “[ ]”, which allows units to be better distinguished from parameter names.

4. The fifth chapter “5. Soaring Methods” is divided into two sub-chapters (“5.1. Static Soaring” and “5.2. Dynamic Soaring”), but there is no guidance or summary of this chapter and the two sub-chapters.

5. Figure 8 (b) lacks a legend.

6. In the title of chapter 4, It would be better to use "Dynamic Model " instead of “Numerical Model”, as written in the last paragraph of this chapter “the developed dynamic model”.

7. In chapter 5 and 6, the description of optimization problem is not concise and standard enough, resulting in the description of optimization problem is not clear in the article. A description similar to the following is recommended:

                                             min f(x)

                      s.t.  gi(x)≤0,   i=1,2,...,m

                             hi(x)=0,   i=1,2,...,m

8. As for the design of aircraft, the article mentioned two concepts, one is aerodynamic design (Chapter 3) and the other is deployment design (Chapter 7), which need to be distinguished. Aerodynamic design, as a separate chapter, needs to be summarized in the abstract or conclusion.
